# Distributionally Robust Neural Networks for Group Shifts: On the Importance of Regularization for Worst-Case Generalization

**Shiori Sagawa**[*]
Stanford University
ssagawa@cs.stanford.edu

**Pang Wei Koh**[*]
Stanford University
pangwei@cs.stanford.edu

**Tatsunori B. Hashimoto**
Microsoft
tahashim@microsoft.com

**Percy Liang**
Stanford University
pliang@cs.stanford.edu

## Abstract

Overparameterized neural networks can be highly accurate *on average* on an i.i.d. test set yet consistently fail on atypical groups of the data (e.g., by learning spurious correlations that hold on average but not in such groups). Distributionally robust optimization (DRO) allows us to learn models that instead minimize the *worst-case* training loss over a set of pre-defined groups. However, we find that naively applying group DRO to overparameterized neural networks fails: these models can perfectly fit the training data, and any model with vanishing average training loss also already has vanishing worst-case training loss. Instead, the poor worst-case performance arises from poor *generalization* on some groups. By coupling group DRO models with increased regularization—a stronger-than-typical $\ell_2$ penalty or early stopping—we achieve substantially higher worst-group accuracies, with 10–40 percentage point improvements on a natural language inference task and two image tasks, while maintaining high average accuracies. Our results suggest that regularization is important for worst-group generalization in the overparameterized regime, even if it is not needed for average generalization. Finally, we introduce a stochastic optimization algorithm, with convergence guarantees, to efficiently train group DRO models.

## 1 Introduction

Machine learning models are typically trained to minimize the average loss on a training set, with the goal of achieving high accuracy on an independent and identically distributed (i.i.d.) test set. However, models that are highly accurate on average can still consistently fail on rare and atypical examples (Hovy & Sgaard, 2015; Blodgett et al., 2016; Tatman, 2017; Hashimoto et al., 2018; Duchi et al., 2019). Such models are problematic when they violate equity considerations (Jurgens et al., 2017; Buolamwini & Gebru, 2018) or rely on *spurious correlations*: misleading heuristics that work for most training examples but do not always hold. For example, in natural language inference (NLI)—determining if two sentences agree or contradict—the presence of negation words like 'never' is strongly correlated with contradiction due to artifacts in crowdsourced training data (Gururangan et al., 2018; McCoy et al., 2019). A model that learns this spurious correlation would be accurate on average on an i.i.d. test set but suffer high error on groups of data where the correlation does not hold (e.g., the group of contradictory sentences with no negation words).

To avoid learning models that rely on spurious correlations and therefore suffer high loss on some groups of data, we instead train models to minimize the *worst-case* loss over groups in the training data. The choice of how to group the training data allows us to use our prior knowledge of spurious correlations, e.g., by grouping together contradictory sentences with no negation words in the NLI example above. This training procedure is an instance of distributionally robust optimization (DRO),

---

[*]Equal contribution.

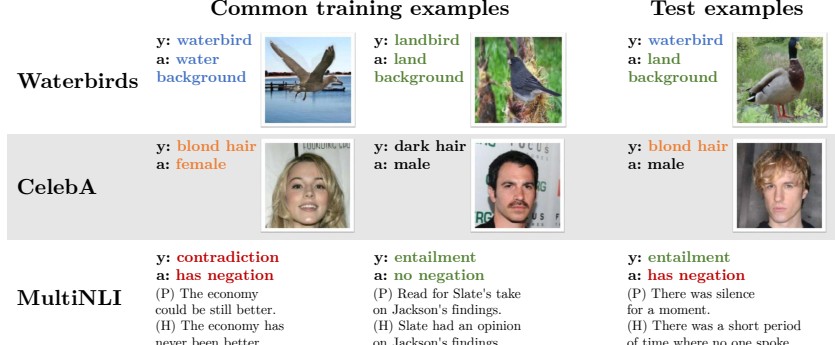

Figure 1: Representative training and test examples for the datasets we consider. The correlation between the label $y$ and the spurious attribute $a$ at training time does not hold at test time.

which optimizes for the worst-case loss over potential test distributions (Ben-Tal et al., 2013; Duchi et al., 2016). Existing work on DRO has focused on models that cannot approach zero training loss, such as generative models (Oren et al., 2019) or convex predictive models with limited capacity (Maurer & Pontil, 2009; Shafieezadeh-Abadeh et al., 2015; Namkoong & Duchi, 2017; Duchi & Namkoong, 2018; Hashimoto et al., 2018).

We study group DRO in the context of overparameterized neural networks in three applications (Figure 1)—natural language inference with the MultiNLI dataset (Williams et al., 2018), facial attribute recognition with CelebA (Liu et al., 2015), and bird photograph recognition with our modified version of the CUB dataset (Wah et al., 2011). The problem with applying DRO to overparameterized models is that if a model achieves zero training loss, then it is optimal on both the worst-case (DRO) and the average training objectives (Zhang et al., 2017; Wen et al., 2014). In the vanishing-training-loss regime, we indeed find that group DRO models do no better than standard models trained to minimize average loss via empirical risk minimization (ERM): both models have high average test accuracies and worst-group *training* accuracies, but low worst-group *test* accuracies (Section 3.1). In other words, the generalization gap is small on average but large for the worst group.

In contrast, we show that strongly-regularized group DRO models that do not attain vanishing training loss can significantly outperform both regularized and unregularized ERM models. We consider $\ell_2$ penalties, early stopping (Section 3.2), and *group adjustments* that minimize a risk measure which accounts for the differences in generalization gaps between groups (Section 3.3). Across the three applications, regularized group DRO improves worst-case test accuracies by 10–40 percentage points while maintaining high average test accuracies. These results give a new perspective on generalization in neural networks: regularization might not be important for good average performance (e.g., models can "train longer and generalize better" on average (Hoffer et al., 2017)) but it appears important for good worst-case performance.

Finally, to carry out the experiments, we introduce a new stochastic optimizer for group DRO that is stable and scales to large models and datasets. We derive convergence guarantees for our algorithm in the convex case and empirically show that it behaves well in our non-convex models (Section 5).

## 2 SETUP

Consider predicting labels $y \in \mathcal{Y}$ from input features $x \in \mathcal{X}$. Given a model family $\Theta$, loss $\ell : \Theta \times (\mathcal{X} \times \mathcal{Y}) \to \mathbb{R}_+$, and training data drawn from some distribution $P$, the standard goal is to find a model $\theta \in \Theta$ that minimizes the expected loss $\mathbb{E}_P[\ell(\theta; (x, y)]$ under the same distribution $P$. The standard training procedure for this goal is empirical risk minimization (ERM):

$$\hat{\theta}_{\text{ERM}} := \arg\min_{\theta \in \Theta} \; \mathbb{E}_{(x,y)\sim\hat{P}}[\ell(\theta; (x, y))], \tag{1}$$

where $\hat{P}$ is the empirical distribution over the training data.

In distributionally robust optimization (DRO) (Ben-Tal et al., 2013; Duchi et al., 2016), we aim instead to minimize the worst-case expected loss over an uncertainty set of distributions $\mathcal{Q}$:

$$\min_{\theta \in \Theta} \left\{ \mathcal{R}(\theta) := \sup_{Q \in \mathcal{Q}} \mathbb{E}_{(x,y)\sim Q}[\ell(\theta; (x, y))] \right\}. \tag{2}$$

The uncertainty set $\mathcal{Q}$ encodes the possible test distributions that we want our model to perform well on. Choosing a general family $\mathcal{Q}$, such as a divergence ball around the training distribution, confers robustness to a wide set of distributional shifts, but can also lead to overly pessimistic models which optimize for implausible worst-case distributions (Duchi et al., 2019).

To construct a realistic set of possible test distributions without being overly conservative, we leverage prior knowledge of spurious correlations to define groups over the training data and then define the uncertainty set $\mathcal{Q}$ in terms of these groups. Concretely, we adopt the *group DRO* setting (Hu et al., 2018; Oren et al., 2019) where the training distribution $P$ is assumed to be a mixture of $m$ groups $P_g$ indexed by $\mathcal{G} = \{1, 2, \ldots, m\}$.[1] We define the uncertainty set $\mathcal{Q}$ as any mixture of these groups, i.e., $\mathcal{Q} := \{\sum_{g=1}^{m} q_g P_g : q \in \Delta_m\}$, where $\Delta_m$ is the $(m-1)$-dimensional probability simplex; this choice of $\mathcal{Q}$ allows us to learn models that are robust to group shifts. Because the optimum of a linear program is attained at a vertex, the worst-case risk (2) is equivalent to a maximum over the expected loss of each group,

$$\mathcal{R}(\theta) = \max_{g \in \mathcal{G}} \mathbb{E}_{(x,y) \sim P_g}[\ell(\theta; (x,y))]. \tag{3}$$

We assume that we know which group each training point comes from—i.e., the training data comprises $(x, y, g)$ triplets—though we do not assume we observe $g$ at test time, so the model cannot use $g$ directly. Instead, we learn a *group DRO* model minimizing the empirical worst-group risk $\hat{\mathcal{R}}(\theta)$:

$$\hat{\theta}_{\text{DRO}} := \arg\min_{\theta \in \Theta} \left\{ \hat{\mathcal{R}}(\theta) := \max_{g \in \mathcal{G}} \mathbb{E}_{(x,y) \sim \hat{P}_g}[\ell(\theta; (x,y))] \right\}, \tag{4}$$

where each group $\hat{P}_g$ is an empirical distribution over all training points $(x, y, g')$ with $g' = g$ (or equivalently, a subset of training examples drawn from $P_g$). Group DRO learns models with good worst-group *training* loss across groups. This need not imply good worst-group *test* loss because of the worst-group *generalization gap* $\delta := \mathcal{R}(\theta) - \hat{\mathcal{R}}(\theta)$. We will show that for overparameterized neural networks, $\delta$ is large unless we apply sufficient regularization.

## 2.1 APPLICATIONS

In the rest of this paper, we study three applications that share a similar structure (Figure 1): each data point $(x, y)$ has some input attribute $a(x) \in \mathcal{A}$ that is spuriously correlated with the label $y$, and we use this prior knowledge to form $m = |\mathcal{A}| \times |\mathcal{Y}|$ groups, one for each value of $(a, y)$. We expect that models that learn the correlation between $a$ and $y$ in the training data would do poorly on groups for which the correlation does not hold and hence do worse on the worst-group loss $\mathcal{R}(\theta)$.

**Object recognition with correlated backgrounds (Waterbirds dataset).** Object recognition models can spuriously rely on the image background instead of learning to recognize the actual object (Ribeiro et al., 2016). We study this by constructing a new dataset, Waterbirds, which combines bird photographs from the Caltech-UCSD Birds-200-2011 (CUB) dataset (Wah et al., 2011) with image backgrounds from the Places dataset (Zhou et al., 2017). We label each bird as one of $\mathcal{Y} = \{\text{waterbird}, \text{landbird}\}$ and place it on one of $\mathcal{A} = \{\text{water background}, \text{land background}\}$, with waterbirds (landbirds) more frequently appearing against a water (land) background (Appendix C.1). There are $n = 4795$ training examples and 56 in the smallest group (waterbirds on land).

**Object recognition with correlated demographics (CelebA dataset).** Object recognition models (and other ML models more generally) can also learn spurious associations between the label and demographic information like gender and ethnicity (Buolamwini & Gebru, 2018). We examine this on the CelebA celebrity face dataset (Liu et al., 2015), using hair color ($\mathcal{Y} = \{\text{blond}, \text{dark}\}$) as the target and gender ($\mathcal{A} = \{\text{male}, \text{female}\}$) as the spurious attribute. There are $n = 162770$ training examples in the CelebA dataset, with 1387 in the smallest group (blond-haired males).

**Natural language inference (MultiNLI dataset).** In natural language inference, the task is to determine if a given hypothesis is entailed by, neutral with, or contradicts a given premise. Prior work has shown that crowdsourced training datasets for this task have significant annotation artifacts, such as the spurious correlation between contradictions and the presence of the negation words

---

[1] In our main experiments, $m = 4$ or 6; we also use $m = 64$ in our supplemental experiments.

*nobody*, *no*, *never*, and *nothing* (Gururangan et al., 2018). We divide the MultiNLI dataset (Williams et al., 2018) into $m = 6$ groups, one for each pair of labels $\mathcal{Y} = \{\text{entailed}, \text{neutral}, \text{contradictory}\}$ and spurious attributes $\mathcal{A} = \{\text{no negation}, \text{negation}\}$. There are $n = 206175$ examples in our training set, with 1521 examples in the smallest group (entailment with negations); see Appendix C.1 for more details on dataset construction and the training/test split.

# 3   COMPARISON BETWEEN GROUP DRO AND ERM

To study the behavior of group DRO vs. ERM in the overparametrized setting, we fine-tuned ResNet50 models (He et al., 2016) on Waterbirds and CelebA and a BERT model (Devlin et al., 2019) on MultiNLI. These are standard models for image classification and natural language inference which achieve high average test accuracies on their respective tasks.

We train the ERM (1) and group DRO (4) models using standard (minibatch) stochastic gradient descent and (minibatch) stochastic algorithm introduced in Section 5, respectively. We tune the learning rate for ERM and use the same setting for DRO (Appendix C.2). For each model, we measure its *average* (in-distribution) accuracy over training and test sets drawn from the same distribution, as well as its *worst-group* accuracy on the worst-performing group.

## 3.1   ERM AND DRO HAVE POOR WORST-GROUP ACCURACY IN THE OVERPARAMETERIZED REGIME

Overparameterized neural networks can perfectly fit the training data and still generalize well on average (Zhang et al., 2017). We start by showing that these overparameterized models do not generalize well on the worst-case group when they are trained to convergence using standard regularization and hyperparameter settings (He et al., 2016; Devlin et al., 2019), regardless of whether they are trained with ERM or group DRO.[2]

**ERM.**   As expected, ERM models attain near-perfect worst-group training accuracies of at least $99.9\%$ on all three datasets and also obtain high average test accuracies ($97.3\%$, $94.8\%$, and $82.5\%$ on Waterbirds, CelebA, and MultiNLI). However, they perform poorly on the worst-case group at test time with worst-group accuracies of $60.0\%$, $41.1\%$, and $65.7\%$ respectively (Table 1, Figure 2). Their low worst-group accuracies imply that these models are brittle under group shifts.

**DRO.**   The ERM models trained above nearly perfectly classify every training point, and are therefore near-optimal for both the ERM (1) and DRO (4) objectives. Indeed, we find that group DRO models perform similarly to ERM models, attaining near-perfect training accuracies and high average test accuracies, but poor worst-group test accuracies (Table 1, Figure 2).

**Discussion.**   The ERM and DRO models attain near-perfect training accuracy and vanishing training loss even in the presence of default regularization (batch normalization and standard $\ell_2$ penalties for ResNet50, and dropout for BERT). However, despite generalizing well on average, they do *not* generalize well on the worst-case group, and consequently suffer from low worst-group accuracies. This gap between average and worst-group test accuracies arises not from poor worst-group training performance—the models are near-perfect at training time, even on the worst-case groups—but from variations in the generalization gaps across groups. Even though DRO is designed to improve worst-group performance, we find no improvements on worst-group test accuracies since the models already achieve vanishing worst-group losses on the *training* data.

## 3.2   DRO IMPROVES WORST-GROUP ACCURACY UNDER APPROPRIATE REGULARIZATION

Classically, we can control the generalization gap with regularization techniques that constrain the model family's capacity to fit the training data. In the modern overparameterized regime, explicit

---

[2] Training to convergence is a widespread practice for image models (Zhang et al., 2017; Hoffer et al., 2017). Pre-trained language models are typically pretrained until convergence (Devlin et al., 2019; Radford et al., 2019) but fine-tuned for a fixed small number of epochs because average test accuracy levels off quickly; we verified that training to convergence gave equally high average test accuracy.

|  |  |  | Average Accuracy | | Worst-Group Accuracy | |
|  |  |  | ERM | DRO | ERM | DRO |
| --- | --- | --- | --- | --- | --- | --- |
| Standard Regularization | Waterbirds | Train | 100.0 | 100.0 | 100.0 | 100.0 |
|  |  | Test | 97.3 | 97.4 | 60.0 | 76.9 |
|  | CelebA | Train | 100.0 | 100.0 | 99.9 | 100.0 |
|  |  | Test | 94.8 | 94.7 | 41.1 | 41.1 |
|  | MultiNLI | Train | 99.9 | 99.3 | 99.9 | 99.0 |
|  |  | Test | 82.5 | 82.0 | 65.7 | 66.4 |
| Strong $\ell_2$ Penalty | Waterbirds | Train | 97.6 | 99.1 | 35.7 | 97.5 |
|  |  | Test | 95.7 | 96.6 | 21.3 | 84.6 |
|  | CelebA | Train | 95.7 | 95.0 | 40.4 | 93.4 |
|  |  | Test | 95.8 | 93.5 | 37.8 | 86.7 |
| Early Stopping | Waterbirds | Train | 86.2 | 80.1 | 7.1 | 74.2 |
|  |  | Test | 93.8 | 93.2 | 6.7 | 86.0 |
|  | CelebA | Train | 91.3 | 87.5 | 14.2 | 85.1 |
|  |  | Test | 94.6 | 91.8 | 25.0 | 88.3 |
|  | MultiNLI | Train | 91.5 | 86.1 | 78.6 | 83.3 |
|  |  | Test | 82.8 | 81.4 | 66.0 | 77.7 |

Table 1: Average and worst-group accuracies for each training method. Both ERM and DRO models perform poorly on the worst-case group in the absence of regularization (top). With strong regularization (middle, bottom), DRO achieves high worst-group performance, significantly improving from ERM. Cells are colored by accuracy, from low (red) to medium (white) to high (blue) accuracy.

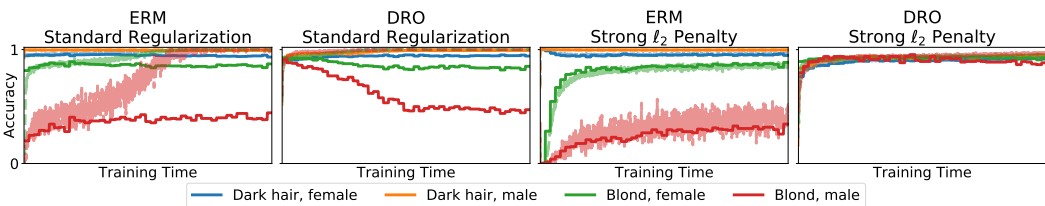

Figure 2: Training (light) and validation (dark) accuracy for CelebA throughout training. With default hyperparameters and training to convergence, ERM and DRO models achieve perfect training accuracy across groups, but generalize badly on the worst-case group (red line in the left panels). With strong $\ell_2$ penalties, ERM models get high average train and test accuracies at the cost of the rare group (panel 3). DRO models achieve high train and test accuracies across groups (panel 4).

regularization is not critical for average performance: models can do well on average even when all regularization is removed (Zhang et al., 2017), and default regularization settings (like in the models trained above) still allow models to perfectly fit the training data. Here, we study if increasing regularization strength—until the models no longer perfectly fit the training data—can rescue worst-case performance. We find that departing from the vanishing-training-loss regime allows DRO models to significantly outperform ERM models on worst-group test accuracy while maintaining high average accuracy. We investigate two types of regularization:

$\ell_2$ **penalties.**  The default coefficient of the $\ell_2$-norm penalty $\lambda\|\theta\|_2^2$ in ResNet50 is $\lambda = 0.0001$ (He et al., 2016). We find that increasing $\lambda$ by several orders of magnitude—to $\lambda = 1.0$ for Waterbirds and $\lambda = 0.1$ for CelebA—does two things: 1) it prevents both ERM and DRO models from achieving perfect training accuracy, and 2) substantially reduces the generalization gap for each group.

With strong $\ell_2$ penalties, both ERM and DRO models still achieve high average test accuracies. However, because no model can achieve perfect training accuracy in this regime, ERM models sacrifice worst-group training accuracy (35.7% and 40.4% on Waterbirds and CelebA; Table 1, Figure 2) and consequently obtain poor worst-group test accuracies (21.3% and 37.8%, respectively).

In contrast, DRO models attain high worst-group training accuracy ($97.5\%$ and $93.4\%$ on Waterbirds and CelebA). The small generalization gap in the strong-$\ell_2$-penalty regime means that high worst-group training accuracy translates to high worst-group test accuracy, which improves over ERM from $21.3\%$ to $84.6\%$ on Waterbirds and from $37.8\%$ to $86.7\%$ on CelebA.

While these results show that strong $\ell_2$ penalties have a striking impact on ResNet50 models for Waterbirds and CelebA, we found that increasing the $\ell_2$ penalty on the BERT model for MultiNLI resulted in similar or worse robust accuracies than the default BERT model with no $\ell_2$ penalty.

**Early stopping.** A different, implicit form of regularization is early stopping (Hardt et al., 2016b). We use the same settings in Section 3.1, but only train each model for a fixed (small) number of epochs (Section C.2). As with strong $\ell_2$ penalties, curtailing training reduces the generalization gap and prevents models from fitting the data perfectly. In this setting, DRO also does substantially better than ERM on worst-group test accuracy, improving from $6.7\%$ to $86.0\%$ on Waterbirds, $25.0\%$ to $88.3\%$ on CelebA, and $66.0\%$ to $77.7\%$ on MultiNLI. Average test accuracies are comparably high in both ERM and DRO models, though there is a small drop of $1-3\%$ for DRO (Table 1, Figure 2).

**Discussion.** We conclude that regularization—preventing the model from perfectly fitting the training data—does matter for worst-group accuracy. Specifically, it controls the generalization gap for each group, even on the worst-case group. Good worst-group test accuracy then becomes a question of good worst-group training accuracy. Since no regularized model can perfectly fit the training data, ERM and DRO models make different training trade-offs: ERM models sacrifice worst-group for average training accuracy and therefore have poor worst-group test accuracies, while DRO models maintain high worst-group training accuracy and therefore do well at test time. Our findings raise questions about the nature of generalization in neural networks, which has been predominantly studied only in the context of average accuracy (Zhang et al., 2017; Hoffer et al., 2017).

### 3.3 ACCOUNTING FOR GENERALIZATION THROUGH GROUP ADJUSTMENTS IMPROVES DRO

In the previous section, we optimized for the worst-group *training* loss via DRO (4), relying on regularization to control the worst-group generalization gap and translate good worst-group training loss to good worst-group test loss. However, even with regularization, the generalization gap can vary significantly across groups: in the Waterbirds DRO model with a strong $\ell_2$ penalty, the smallest group has a train-test accuracy gap of $15.4\%$ compared to just $1.0\%$ for the largest group. This suggests that we can obtain better worst-group test loss if at training time, we prioritize obtaining lower training loss on the groups that we expect to have a larger generalization gap.

We make this approach concrete by directly minimizing an estimated upper bound on the worst-group test loss, inspired by ideas from structural risk minimization (Vapnik, 1992). The key consideration is that each group $g$ has its own generalization gap $\delta_g = \mathbb{E}_{(x,y)\sim P_g}[\ell(\theta;(x,y))] - \mathbb{E}_{(x,y)\sim \hat{P}_g}[\ell(\theta;(x,y))]$. To approximate optimizing for the worst-group test loss $\mathcal{R}(\theta) = \max_{g\in\mathcal{G}}\mathbb{E}_{(x,y)\sim \hat{P}_g}[\ell(\theta;(x,y))] + \delta_g$, we propose using the simple, parameter-independent heuristic $\hat{\delta}_g = C/\sqrt{n_g}$, where $n_g$ is the group size for $g$ and $C$ is a model capacity constant which we treat as a hyperparameter. This gives the *group-adjusted* DRO estimator

$$\hat{\theta}_{\text{adj}} := \underset{\theta\in\Theta}{\arg\min}\ \max_{g\in\mathcal{G}}\left\{\mathbb{E}_{(x,y)\sim \hat{P}_g}[\ell(\theta;(x,y))] + \frac{C}{\sqrt{n_g}}\right\}. \tag{5}$$

The scaling with $1/\sqrt{n_g}$ reflects how smaller groups are more prone to overfitting than larger groups, and is inspired by the general size dependence of model-complexity-based generalization bounds (see, e.g., Cao et al. (2019)).

By incorporating group adjustments in (5), we encourage the model to focus more on fitting the smaller groups. We note that this method of using a $1/\sqrt{n}$ surrogate for the generalization gap only works in the group DRO setting, where we consider the worst-group loss over groups of different sizes. It does not apply in the ERM setting; if we were minimizing average training loss, the $1/\sqrt{n}$ term would simply be a constant and not affect the optimization.

**Results.** We evaluate group adjustments using group DRO models with strong $\ell_2$ penalties (as in Section 3.2). In Waterbirds ($\lambda = 1.0$), worst-group test accuracy improves by $5.9\%$, cutting the

| | Average Accuracy | | Worst-Group Accuracy | |
|---|---|---|---|---|
| | Naïve | Adjusted | Naïve | Adjusted |
| Waterbirds | 96.6 | 93.7 | 84.6 | 90.5 |
| CelebA | 93.5 | 93.4 | 86.7 | 87.8 |

Table 2: Average and worst-group test accuracies with and without group adjustments. Group adjustments improve worst-group accuracy, though average accuracy drops for Waterbirds.

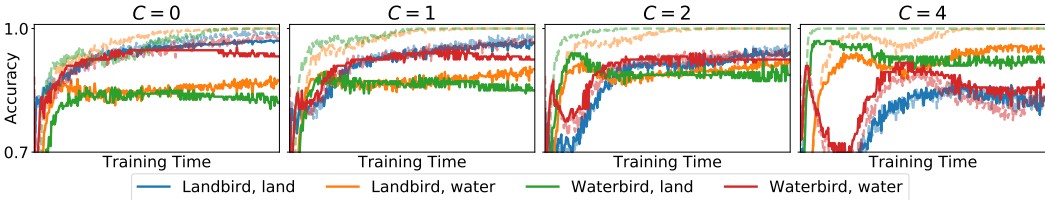

Figure 3: Training (light) and validation (dark) accuracies for each group over time, for different adjustments $C$. When $C = 0$, the generalization gap for waterbirds on land (green line) is large, dragging down worst-group accuracy. At $C = 2$, which has the best worst-group validation accuracy, the accuracies are balanced. At $C = 4$, we overcompensate for group sizes, so smaller groups (e.g., waterbirds on land) do better at the expense of larger groups (e.g., landbirds on land).

error rate by more than a third (Table 2 and Figure 3). The improvements in CelebA ($\lambda = 0.1$) are more modest, with worst-group accuracy increasing by $1.1\%$; $\ell_2$ penalties are more effective in CelebA and there is not as much variation in the generalization gaps by group at $\lambda = 0.1$. We did not evaluate group adjustments on MultiNLI as it did not benefit from stronger $\ell_2$ penalties.

Empirically, group adjustments also help in the early stopping setting of Section 3.2 (in the next section, we evaluate models with group adjustments and early stopping across a grid of $\ell_2$ penalty strengths). However, it is difficult to rigorously study the effects of early stopping (e.g., because the group losses have not converged to a stable value), so we leave a more thorough investigation of the interaction between early stopping and group adjustments to future work.

## 4 COMPARISON BETWEEN DRO AND IMPORTANCE WEIGHTING

Our results above show that strongly-regularized DRO models can be significantly more robust than ERM models. Here, we show theoretically and empirically that DRO also outperforms a strong importance weighting baseline that is commonly used in machine learning tasks where the train and test distributions differ (Shimodaira, 2000; Byrd & Lipton, 2019). Recall that in our setting, the test distribution can be any mixture of the group distributions. For some assignment of weights $w \in \Delta_m$ to groups, an importance-weighted estimator would learn

$$\hat{\theta}_w := \underset{\theta \in \Theta}{\arg\min} \ \mathbb{E}_{(x,y,g)\sim\hat{P}}[w_g\, \ell(\theta; (x,y))]. \tag{6}$$

**Empirical comparison.** We consider an importance-weighted baseline with weights set to the inverse training frequency of each group, $w_g = 1/\mathbb{E}_{g'\sim\hat{P}}[\mathbb{I}(g' = g)]$. This optimizes for a test distribution with uniform group frequencies and is analogous to the common upweighting technique for label shifts (Cui et al., 2019; Cao et al., 2019); intuitively, this attempts to equalize average and worst-group error by upweighting the minority groups. Concretely, we train our weighted model by sampling from each group with equal probability (Shen et al., 2016), since a recent study found this to be more effective than similar reweighting/resampling methods (Buda et al., 2018).

Unlike group DRO, upweighting the minority groups does not necessarily yield uniformly low training losses across groups in practice, as some groups might be easier to fit than others. To compare upweighting (UW) with ERM and DRO, we train models across the same grid of $\ell_2$ penalty strengths and early stopping at the epoch with best worst-group validation accuracy (Table 3).[3] In CelebA and

---

[3]To avoid advantaging the DRO models by allowing them to tune additional hyperparameters, we restrict our search for group adjustments to the one $\ell_2$ penalty strength used in Section 3.3. See Appendix C.2.

Waterbirds, upweighting performs much better than ERM but is slightly outperformed by DRO. However, upweighting fails on MultiNLI, achieving lower average and worst-group accuracies than even ERM. With upweighting, it appears that the rare group is overemphasized and extremely low training accuracy is achieved for that group at the cost of others.

| | Average Accuracy | | | Worst-Group Accuracy | | |
|---|---|---|---|---|---|---|
| | ERM | UW | DRO | ERM | UW | DRO |
| Waterbids | **97.0 (0.2)** | 95.1 (0.3) | 93.5 (0.3) | 63.7 (1.9) | 88.0 (1.3) | **91.4 (1.1)** |
| CelebA | **94.9 (0.2)** | 92.9 (0.2) | 92.9 (0.2) | 47.8 (3.7) | 83.3 (2.8) | **88.9 (2.3)** |
| MultiNLI | **82.8 (0.1)** | 81.2 (0.1) | 81.4 (0.1) | 66.4 (1.6) | 64.8 (1.6) | **77.7 (1.4)** |

Table 3: Comparison of ERM, upweighting (UW), and group DRO models, with binomial standard deviation in parenthesis. For each objective, we grid search over $\ell_2$ penalty strength, number of epochs, and group adjustments and report on the model with highest validation accuracy. These numbers differ from the previous tables because of the larger grid search.

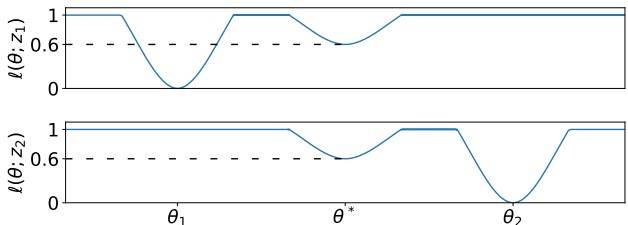

Figure 4: Toy example illustrating that DRO and importance weighting are not equivalent. The DRO solution is $\theta^*$, while any importance weighting would result in solutions at $\theta_1$ or $\theta_2$.

**Theoretical comparison.** Should we expect importance weighting to learn models with good worst-case loss? We show that importance weighting and DRO can learn equivalent models in the convex setting under some importance weights, but not necessarily when the models are non-convex.

We analyze the general framework of having weights $w(z)$ for each data point $z$, which is more powerful than the specific choice above of assigning weights by groups. By minimizing the weighted loss $\mathbb{E}_{z \sim P}[w(z)\ell(\theta; z)]$ over some source distribution $P$, we can equivalently minimize the expected loss $\mathbb{E}_{z \sim Q}[\ell(\theta; z)]$ over a target distribution $Q$ where $Q(z) \propto w(z)P(z)$. However, we want good worst-case performance over a family of $Q \in \mathcal{Q}$, instead of a single $Q$. Are there weights $w$ such that the resulting model $\hat{\theta}_w$ achieves optimal worst-group risk? In the convex regime, standard duality arguments show that this is the case (see Appendix A.1 for the proof):

**Proposition 1.** *Suppose that the loss $\ell(\cdot; z)$ is continuous and convex for all $z$ in $\mathcal{Z}$, and let the uncertainty set $\mathcal{Q}$ be a set of distributions supported on $\mathcal{Z}$. Assume that $\mathcal{Q}$ and the model family $\Theta \subseteq \mathbb{R}^d$ are convex and compact, and let $\theta^* \in \Theta$ be a minimizer of the worst-group objective $\mathcal{R}(\theta)$. Then there exists a distribution $Q^* \in \mathcal{Q}$ such that $\theta^* \in \arg\min_\theta \mathbb{E}_{z \sim Q^*}[\ell(\theta; z)]$.*

However, this equivalence breaks down when the loss $\ell$ is non-convex:

**Counterexample 1.** *Consider a uniform data distribution $P$ supported on two points $\mathcal{Z} = \{z_1, z_2\}$, and let $\ell(\theta; z)$ be as in Figure 4, with $\Theta = [0, 1]$. The DRO solution $\theta^*$ achieves a worst-case loss of $\mathcal{R}(\theta^*) = 0.6$. Now consider any weights $(w_1, w_2) \in \Delta_2$ and w.l.o.g. let $w_1 \geq w_2$. The minimizer of the weighted loss $w_1 \ell(\theta; z_1) + w_2 \ell(\theta; z_2)$ is $\theta_1$, which only attains a worst-case loss of $\mathcal{R}(\theta^*) = 1.0$.*

**Remark.** *Under regularity conditions, there exists a distribution $Q$ such that $\theta^*$ is a first-order stationary point of $\mathbb{E}_{z \sim Q}[\ell(\theta; z)]$ (see e.g., Arjovsky et al. (2019)). However, as the counterexample demonstrates, in the non-convex setting this does not imply that $\theta^*$ actually minimizes $\mathbb{E}_{z \sim Q}[\ell(\theta; z)]$.*

This negative result implies that in the non-convex setting, there may not be *any* choice of weights $w$ such that the resulting minimizer $\hat{\theta}_w$ is robust. Even if such weights did exist, they depend on $\theta^*$ and obtaining these weights requires that we solve a dual DRO problem, making reweighting no easier to implement than DRO. Common choices of weights, such as inverse group size, are heuristics that may not yield robust solutions (as observed for MultiNLI in Table 3).

## 5 ALGORITHM

To train group DRO models efficiently, we introduce an online optimization algorithm with convergence guarantees. Prior work on group DRO has either used batch optimization algorithms, which do not scale to large datasets, or stochastic optimization algorithms without convergence guarantees.

In the convex and batch case, there is a rich literature on distributionally robust optimization which treats the problem as a standard convex conic program (Ben-Tal et al., 2013; Duchi et al., 2016; Bertsimas et al., 2018; Lam & Zhou, 2015). For general non-convex DRO problems, two types of stochastic optimization methods have been proposed: (i) stochastic gradient descent (SGD) on the Lagrangian dual of the objective (Duchi & Namkoong, 2018; Hashimoto et al., 2018), and (ii) direct minimax optimization (Namkoong & Duchi, 2016). The first approach fails for group DRO because the gradient of the dual objective is difficult to estimate in a stochastic and unbiased manner.[4] An algorithm of the second type has been proposed for group DRO (Oren et al., 2019), but this work does not provide convergence guarantees, and we observed instability in practice under some settings.

Recall that we aim to solve the optimization problem (4), which can be rewritten as

$$\min_{\theta \in \Theta} \sup_{q \in \Delta_m} \sum_{g=1}^{m} q_g \mathbb{E}_{(x,y) \sim P_g}[\ell(\theta; (x,y))]. \tag{7}$$

Extending existing minimax algorithms for DRO (Namkoong & Duchi, 2016; Oren et al., 2019), we interleave gradient-based updates on $\theta$ and $q$. Intuitively, we maintain a distribution $q$ over groups, with high masses on high-loss groups, and update on each example proportionally to the mass on its group. Concretely, we interleave SGD on $\theta$ and exponentiated gradient ascent on $q$ (Algorithm 1). (In practice, we use minibatches and a momentum term for $\theta$; see Appendix C.2 for details.) The key improvement from the existing group DRO algorithm (Oren et al., 2019) is that $q$ is updated using gradients instead of picking the group with worst average loss at each iteration, which is important for stability and obtaining convergence guarantees. The run time of the algorithm is similar to that of SGD for a given number of epochs (less than a 5% difference), as run time is dominated by the computation of the loss and its gradient.

---

**Algorithm 1:** Online optimization algorithm for group DRO

---

**Input:** Step sizes $\eta_q, \eta_\theta; P_g$ for each $g \in \mathcal{G}$
Initialize $\theta^{(0)}$ and $q^{(0)}$
**for** $t = 1, \ldots, T$ **do**
$\quad g \sim \text{Uniform}(1, \ldots, m)$          // Choose a group $g$ at random
$\quad x, y \sim P_g$                  // Sample $x, y$ from group $g$
$\quad q' \leftarrow q^{(t-1)}; q'_g \leftarrow q'_g \exp(\eta_q \ell(\theta^{(t-1)}; (x,y)))$    // Update weights for group $g$
$\quad q^{(t)} \leftarrow q' / \sum_{g'} q'_{g'}$                    // Renormalize $q$
$\quad \theta^{(t)} \leftarrow \theta^{(t-1)} - \eta_\theta q_g^{(t)} \nabla \ell(\theta^{(t-1)}; (x,y))$      // Use $q$ to update $\theta$
**end**

---

We analyze the convergence rate by studying the error $\varepsilon_T$ of the average iterate $\bar{\theta}^{(1:T)}$:

$$\varepsilon_T = \max_{q \in \Delta_m} L(\bar{\theta}^{(1:T)}, q) - \min_{\theta \in \Theta} \max_{q \in \Delta_m} L(\theta, q), \tag{8}$$

where $L(\theta, q) := \sum_{g=1}^{m} q_g \mathbb{E}_{(x,y) \sim P_g}[\ell(\theta; (x,y))]$ is the expected worst-case loss. Applying results from Nemirovski et al. (2009), we can show that Algorithm 1 has a standard convergence rate of $O(1/\sqrt{T})$ in the convex setting (proof in Section A.2):

**Proposition 2.** *Suppose that the loss* $\ell(\cdot; (x,y))$ *is non-negative, convex,* $B_\nabla$*-Lipschitz continuous, and bounded by* $B_\ell$ *for all* $(x,y)$ *in* $\mathcal{X} \times \mathcal{Y}$*, and* $\|\theta\|_2 \leq B_\Theta$ *for all* $\theta \in \Theta$ *with convex* $\Theta \subseteq \mathbb{R}^d$*. Then, the average iterate of Algorithm 1 achieves an expected error at the rate*

$$\mathbb{E}[\varepsilon_T] \leq 2m \sqrt{\frac{10(B_\Theta^2 B_\nabla^2 + B_\ell^2 \log m)}{T}}. \tag{9}$$

*where the expectation is taken over the randomness of the algorithm.*

---

[4] The dual optimization problem for group DRO is $\min_{\theta, \beta} \frac{1}{\alpha} \mathbb{E}_g[\max(0, \mathbb{E}_{x,y \sim \hat{P}_g}[\ell(\theta; (x,y)) \mid g] - \beta)] + \beta$ for constant $\alpha$. The max over *expected* loss makes it difficult to obtain an unbiased, stochastic gradient estimate.

## 6 RELATED WORK

**The problem of non-uniform accuracy.**  Existing approaches to addressing non-uniform accuracy over the data distribution include domain adaptation techniques for *known* target distributions (Ben-David et al., 2006; Ganin & Lempitsky, 2015) and work in ML fairness (Dwork et al., 2012; Hardt et al., 2016a; Kleinberg et al., 2017). As we discuss in Section 4, importance weighting is a classic example of the former (Shimodaira, 2000). Byrd & Lipton (2019) empirically study importance weighting in neural networks and demonstrate that it has little effect unless regularization is applied. This is consistent with the theoretical analysis in Wen et al. (2014), which points out that weighting has little impact in the zero-loss regime, and with our own observations in the context of DRO.

**Distributionally robust optimization.**  Prior work in DRO typically defines the uncertainty set $\mathcal{Q}$ as a divergence ball around the training distribution over $(x, y)$ (Ben-Tal et al., 2013; Lam & Zhou, 2015; Duchi et al., 2016; Miyato et al., 2018; Esfahani & Kuhn, 2018; Bertsimas et al., 2018; Blanchet & Murthy, 2019). With small divergence balls of radii $O(1/n)$, DRO acts as a regularizer (Shafieezadeh-Abadeh et al., 2015; Namkoong & Duchi, 2017). However, when the radius is larger, the resulting $\mathcal{Q}$ can be too pessimistic. In contrast, group DRO considers $\mathcal{Q}$ that is of wider radius but with fewer degrees of freedom (shifts over groups instead of over $(x, y)$). Prior work proposed group DRO in the context of label shifts (Hu et al., 2018) and shifts in data sources (Oren et al., 2019). Our work studies group DRO in the overparameterized regime with vanishing training loss and poor worst-case generalization. In contrast, most DRO work has focused on the classic (underparameterized) model setting (Namkoong & Duchi, 2017; Hu et al., 2018; Duchi et al., 2019). Sinha et al. (2018) study neural networks but with a more conservative Wasserstein uncertainty set that leads to non-vanishing training loss; and Oren et al. (2019) study neural networks but for generative modeling where loss tradeoffs arise naturally.

**Generalization of robust models.**  There is extensive work investigating generalization of neural networks in terms of average loss, theoretically and empirically (Hardt et al., 2016b; Szegedy et al., 2016; Hoffer et al., 2017). However, analysis on robust losses is limited. For label shifts, prior work has observed overfitting on rare labels and proposed algorithms to mitigate it (Buda et al., 2018; Cui et al., 2019; Cao et al., 2019). In the DRO literature, generalization bounds on the DRO objective exist for particular uncertainty sets (e.g., Duchi & Namkoong (2018)), but those works do not study overparameterized models. Invariant prediction models, mostly from the causal inference literature, similarly aim to achieve high performance on a range of test distributions (Peters et al., 2016; Bühlmann & Meinshausen, 2016; Heinze-Deml & Meinshausen, 2017; Rothenhäusler et al., 2018; Yang et al., 2019; Arjovsky et al., 2019). For example, the maximin regression framework (Meinshausen & Bühlmann, 2015) also assumes group-based shifts, but focuses on settings without the generalization problems identified in our work.

## 7 DISCUSSION

In this paper, we analyzed group DRO in overparameterized neural networks and highlighted the importance of regularization for worst-case group generalization. When strongly regularized, group DRO significantly improves worst-group accuracy at a small cost in average accuracy.

As an application, we showed that group DRO can prevent models from learning pre-specified spurious correlations. Our supplemental experiments also suggest that group DRO models can maintain high worst-group accuracy even when groups are imperfectly specified (Appendix B). While handling shifts beyond pre-specified group shifts is important future work, existing work has identified many distributional shifts that can be expressed with pre-specified groups, e.g., batch effects in biology (Leek et al., 2010), or image artifacts (Oakden-Rayner et al., 2019) and patient demographics (Badgeley et al., 2019) in medicine.

More generally, our observations call for a deeper analysis of average vs. worst-case generalization in the overparameterized regime. Such analysis may shed light on the failure modes of deep neural networks as well as provide additional tools (beyond strong $\ell_2$ penalties or early stopping) to counter poor worst-case generalization while maintaining high average accuracy.

## ACKNOWLEDGMENTS

We are grateful to Shyamal Buch, Yair Carmon, Zhenghao Chen, John Duchi, Jean Feng, Christina Heinze-Deml, Robin Jia, Daphne Koller, Ananya Kumar, Tengyu Ma, Jesse Mu, Hongseok Namkoong, Emma Pierson, and Fanny Yang for helpful discussions and suggestions. This work was funded by an Open Philanthropy Project Award. Toyota Research Institute ("TRI") also provided funds to assist the authors with their research but this article solely reflects the opinions and conclusions of its authors and not TRI or any other Toyota entity. SS was supported by a Stanford Graduate Fellowship and PWK was supported by the Facebook Fellowship Program.

## REPRODUCIBILITY

Code for training group DRO models is available at `https://github.com/kohpangwei/group_DRO`. The datasets used in this paper are also available at that link, as well as scripts to modify dataset generation (e.g., to choose different spurious attributes for CelebA and MultiNLI, or different object backgrounds or relative group sizes for Waterbirds).

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

# A PROOFS

## A.1 EQUIVALENCE OF DRO AND IMPORTANCE WEIGHTING IN THE CONVEX SETTING

**Proposition 1.** *Suppose that the loss $\ell(\cdot; z)$ is continuous and convex for all $z$ in $\mathcal{Z}$, and let the uncertainty set $\mathcal{Q}$ be a set of distributions supported on $\mathcal{Z}$. Assume that $\mathcal{Q}$ and the model family $\Theta \subseteq \mathbb{R}^d$ are convex and compact, and let $\theta^* \in \Theta$ be a minimizer of the worst-group objective $\mathcal{R}(\theta)$. Then there exists a distribution $Q^* \in \mathcal{Q}$ such that $\theta^* \in \arg\min_\theta \mathbb{E}_{z \sim Q^*}[\ell(\theta; z)]$.*

*Proof.* Let $h(\theta, Q) := \mathbb{E}_{z \sim Q}[\ell(\theta; z)]$. Since the loss $\ell(\theta; z)$ is continuous and convex in $\theta$ for all $z$ in $\mathcal{Z}$, we have that $h(\theta, Q)$ is continuous, convex in $\theta$, and concave (linear) in $Q$. Moreover, since convexity and lower semi-continuity are preserved under arbitrary pointwise suprema, $\sup_{Q \in \mathcal{Q}} h(\theta, Q)$ is also convex and lower semi-continuous (therefore proper).

Together with the compactness of $\Theta$ and $\mathcal{Q}$, the above conditions imply (by Weierstrass' theorem, proposition 3.2.1, Bertsekas (2009)), that the optimal value of the DRO objective

$$\inf_{\theta \in \Theta} \mathcal{R}(\theta) = \inf_{\theta \in \Theta} \sup_{Q \in \mathcal{Q}} h(\theta, Q). \tag{10}$$

is attained at some $\theta^* \in \Theta$.

A similar argument implies that the sup-inf objective

$$\sup_{Q \in \mathcal{Q}} \inf_{\theta \in \Theta} h(\theta, Q) \tag{11}$$

attains its optimum at some $Q^* \in \mathcal{Q}$.

Moreover, because $\Theta$ and $\mathcal{Q}$ are compact and $h$ is continuous, we have the max-min equality (see, e.g., Ex 5.25 in Boyd & Vandenberghe (2004))

$$\sup_{Q \in \mathcal{Q}} \inf_{\theta \in \Theta} h(\theta, Q) = \inf_{\theta \in \Theta} \sup_{Q \in \mathcal{Q}} h(\theta, Q). \tag{12}$$

Together, the above results imply that $(\theta^*, Q^*)$ form a saddle point (proposition 3.4.1, Bertsekas (2009)), that is,

$$\sup_{Q \in \mathcal{Q}} h(\theta^*, Q) = h(\theta^*, Q^*) = \inf_{\theta \in \Theta} h(\theta, Q^*). \tag{13}$$

In particular, the second equality indicates that the optimal DRO model $\theta^*$ also minimizes the weighted risk $h(\theta, Q^*) = \mathbb{E}_{Z \sim Q^*}[\ell(\theta; Z)]$, as desired.

$\square$

## A.2  CONVERGENCE RATE OF ALGORITHM 1

**Proposition 2.** *Suppose that the loss $\ell(\cdot; (x, y))$ is non-negative, convex, $B_\nabla$-Lipschitz continuous, and bounded by $B_\ell$ for all $(x, y)$ in $\mathcal{X} \times \mathcal{Y}$, and $\|\theta\|_2 \leq B_\Theta$ for all $\theta \in \Theta$ with convex $\Theta \subseteq \mathbb{R}^d$. Then, the average iterate of Algorithm 1 achieves an expected error at the rate*

$$\mathbb{E}[\varepsilon_T] \leq 2m \sqrt{\frac{10[B_\Theta^2 B_\nabla^2 + B_\ell^2 \log m]}{T}}. \tag{14}$$

*where the expectation is taken over the randomness of the algorithm.*

*Proof.* Our proof is an application of the regret bound for online mirror descent on saddle point optimization from Nemirovski et al. (2009).

We first introduce the existing theorem. Consider the saddle-point optimization problem

$$\min_{\theta \in \Theta} \max_{q \in \Delta_m} \sum_{g=1}^{m} q_g f_g(\theta) \tag{15}$$

under the following assumptions:

**Assumption 1.** *$f_g$ is convex on $\Theta$.*

**Assumption 2.** *$f_g(\theta) = \mathbb{E}_{\xi \sim q}[F_g(\theta; \xi)]$ for some function $F_g$.*

**Assumption 3.** *We generate i.i.d. examples $\xi \sim q$. For a given $\theta \in \Theta$ and $\xi \in \Xi$, we can compute $F_g(\theta, \xi)$ and unbiased stochastic subgradient $\nabla F_g(\theta; \xi)$, that is, $\mathbb{E}_{\xi \sim q}[\nabla F_g(\theta; \xi)] = \nabla f_g(\theta)$.*

Online mirror descent with some $c$-strongly convex norm $\| \cdot \|_\theta$, yielding iterates $\theta^{(1)}, \ldots, \theta^{(T)}$ and $q^{(1)}, \ldots, q^{(T)}$, has the following guarantee.

**Theorem 1** (Nemirovski et al. (2009), Eq 3.23). *Suppose that Assumptions 1-3 hold. Then the pseudo-regret of the average iterates $\bar{q}_g^{(1:T)}$ and $\bar{q}_g^{(1:T)}$ can be bounded as*

$$\mathbb{E}\left[\max_{q \in \Delta_m} \sum_{g=1}^{m} q_g f_g(\bar{\theta}^{(1:T)}) - \min_{\theta \in \Theta} \sum_{g=1}^{m} \bar{q}_g^{(1:T)} f_g(\theta)\right] \leq 2\sqrt{\frac{10[R_\theta^2 M_{*,\theta}^2 + M_{*,q}^2 \log m]}{T}}, \tag{16}$$

*where*

$$\mathbb{E}\left[\left\|\nabla_\theta \sum_{g=1}^m qF_g(\theta;\xi)\right\|_{*,\theta}^2\right] \leq M_{*,\theta} \tag{17}$$

$$\mathbb{E}\left[\left\|\nabla_q \sum_{g=1}^m qF_g(\theta;\xi)\right\|_{*,q}^2\right] \leq M_{*,q} \tag{18}$$

$$R_\theta^2 = \frac{1}{c}(\max_\theta \|\theta\|_\theta^2 - \min_\theta \|\theta\|_\theta^2) \tag{19}$$

*for online mirror descent with c-strongly convex norm $\|\cdot\|_\theta$.*

It remains to formulate our algorithm as an instance of online mirror descent applied to the saddle-point problem above. We start by defining the following:

**Definition 1.** *Let $q$ be a distribution over $\xi = (x,y,g)$ that is a uniform mixture of individual group distributions $P_g$:*

$$(x,y,g) \sim q := \frac{1}{m}\sum_{g'=1}^m P_{g'}. \tag{20}$$

**Definition 2.** *Let $F_{g'}(\theta;(x,y,g))) := m\mathbb{I}[g = g']\ell(\theta;(x,y))$. Correspondingly, let $f_{g'} := \mathbb{E}_{P_{g'}}[\ell(\theta;(x,y))]$.*

We now check that Assumptions 1-3 hold under the original assumptions in the statement of Theorem 2:

1. We assume that the loss $\ell(\cdot;(x,y))$ is non-negative, continuous, and convex for all $(x,y)$ in $\mathcal{X} \times \mathcal{Y}$. As a result, $f_g(\theta)$ is non-negative, continuous, and convex on $\Theta$.

2. The expected value of $F_g(\theta)$ over distribution $q$ is $f_g(\theta)$:

$$\begin{aligned}
\mathbb{E}_{x,y,g\sim q}[F_{g'}(\theta;(x,y,g))] &= \frac{1}{m}\sum_{i=1}^m \mathbb{E}_{P_i}\left[F_{g'}(\theta;(x,y,g)) \mid g = i\right] \\
&= \frac{1}{m}\mathbb{E}_{P_{g'}}\left[F_{g'}(\theta;(x,y,g)) \mid g = g'\right] \\
&= \frac{1}{m}\mathbb{E}_{P_{g'}}\left[m\ell(\theta;x,y) \mid g = g'\right] \\
&= \mathbb{E}_{P_{g'}}\left[\ell(\theta;x,y) \mid g = g'\right] \\
&= f_{g'}(\theta).
\end{aligned}$$

3. We can compute an unbiased stochastic subgradient $\nabla F_{g'}(\theta;(x,y,g))$

$$\begin{aligned}
\mathbb{E}_{x,y,g\sim q}[\nabla F_{g'}(\theta;(x,y,g))] &= \mathbb{E}_{x,y,g\sim q}[\nabla m\mathbb{I}[g = g']\ell(\theta;(x,y))] \\
&= \frac{1}{m}\sum_{i=1}^m \mathbb{E}_{P_i}[\nabla m\mathbb{I}[g = g']\ell(\theta;x,y)] \\
&= \mathbb{E}_{Q_{g'}}[\nabla\ell(\theta;(x,y))] \\
&= \nabla f_g(\theta).
\end{aligned}$$

Finally, we compute the constants required for the regret bound in Theorem 1. Recalling the original assumptions of Theorem 2,

1. Bounded losses: $\ell(\theta;(x,y)) \leq B_\ell$ for all $x,y,\theta$

2. Bounded gradients: $\|\nabla\ell(\theta;(x,y))\|_2 \leq B_\nabla$ for all $\theta,x,y$

3. Bounded parameter norm: $\|\theta\|_2 \leq B_\Theta$ for all $\theta \in \Theta$,

we obtain:

$$\mathbb{E}\left[\left\|\nabla_\theta \sum_{g'=1}^m q_{g'} F_{g'}(\theta; (x, y, g))\right\|_{*,\theta}^2\right] \leq m^2 B_\nabla^2 = M_{*,\theta} \tag{21}$$

$$\mathbb{E}\left[\left\|\nabla_q \sum_{g'=1}^m q_{g'} F_{g'}(\theta; (x, y, g))\right\|_{*,q}^2\right] \leq m^2 B_\ell^2 = M_{*,q} \tag{22}$$

$$R_\theta^2 = \max_\theta \|\theta\|_\theta^2 - \min_\theta \|\theta\|_\theta^2 = B_\Theta^2. \tag{23}$$

Plugging in these constants into the regret bound from Theorem 1, we obtain

$$\mathbb{E}\left[\max_{q \in \Delta_m} \sum_{g=1}^m q_g f_g(\bar{\theta}^{(1:T)}) - \min_{\theta \in \Theta} \sum_{g=1}^m \bar{q}_g^{(1:T)} f_g(\theta)\right] \leq 2m\sqrt{\frac{10[B_\Theta^2 B_\nabla^2 + B_\ell^2 \log m]}{T}} \tag{24}$$

This implies Theorem 2 because the minimax game is convex-concave. $\qquad\square$

## B    SUPPLEMENTARY EXPERIMENTS

Group DRO can maintain high robust accuracy even when spurious attributes are not perfectly specified. We repeat the CelebA experiment on models with strong $\ell_2$ penalties (Section 3.2) but with inexact group specifications:

1. Instead of the ground-truth spurious attribute *Male*, we provide a related attribute *Wearing Lipstick*, and

2. We also specify four distractor/non-spurious attributes (*Eyeglasses*, *Smiling*, *Double Chin*, and *Oval Face*).

Optimizing for worst-case performance over all $2^6 = 64$ groups (for all combinations of 5 attributes and 1 label), the DRO model attains $78.9\%$ robust accuracy across the 4 original groups (dark-haired males and females, and blond males and females). These robust accuracies are not far off from the original DRO model with just the ground-truth spurious attribute ($86.7\%$) and significantly outperform the ERM model ($37.8\%$).

## C    EXPERIMENTAL DETAILS

### C.1    DATASETS

**MultiNLI.**    The standard MultiNLI train-test split allocates most examples (approximately $90\%$) to the training set, with another $5\%$ as a publicly-available development set and the last $5\%$ as a held-out test set that is only accessible through online competition leaderboards (Williams et al., 2018). Because we are unable to assess model accuracy on each group through the online leaderboards, we create our own validation and test sets by combining the training set and development set and then randomly shuffling them into a $50-20-30$ train-val-test split. We chose to allocates more examples to the validation and test sets than the standard split to allow us to accurately estimate performance on rare groups in the validation and test sets.

We use the provided gold labels as the target, removing examples with no consensus gold label (as is standard procedure). We annotate an example as having a negation word if any of the words *nobody*, *no*, *never*, and *nothing* appear in the hypothesis (Gururangan et al., 2018).

**Waterbirds.** The CUB dataset (Wah et al., 2011) contains photographs of birds annotated by species as well as and pixel-level segmentation masks of each bird. To construct the Waterbirds dataset, we label each bird as a *waterbird* if it is a seabird (albatross, auklet, cormorant, frigate-bird, fulmar, gull, jaeger, kittiwake, pelican, puffin, or tern) or waterfowl (gadwall, grebe, mallard, merganser, guillemot, or Pacific loon). Otherwise, we label it as a *landbird*.

To control the image background, we use the provided pixel-level segmentation masks to crop each bird out from its original background and onto a water background (categories: *ocean* or *natural lake*) or land background (categories: *bamboo forest* or *broadleaf forest*) obtained from the Places dataset (Zhou et al., 2017). In the training set, we place 95% of all waterbirds against a water background and the remaining 5% against a land background. Similarly, 95% of all landbirds are placed against a land background with the remaining 5% against water.

We refer to this combined CUB-Places dataset as the Waterbirds dataset to avoid confusion with the original fine-grained species classification task in the CUB dataset.

We use the official train-test split of the CUB dataset, randomly choosing 20% of the training data to serve as a validation set. For the validation and test sets, we allocate distribute landbirds and waterbirds equally to land and water backgrounds (i.e., there are the same number of landbirds on land vs. water backgrounds, and separately, the same number of waterbirds on land vs. water backgrounds). This allows us to more accurately measure the performance of the rare groups, and it is particularly important for the Waterbirds dataset because of its relatively small size; otherwise, the smaller groups (waterbirds on land and landbirds on water) would have too few samples to accurately estimate performance on. We note that we can only do this for the Waterbirds dataset because we control the generation process; for the other datasets, we cannot generate more samples from the rare groups.

In a typical application, the validation set might be constructed by randomly dividing up the available training data. We emphasize that this is not the case here: the training set is skewed, whereas the validation set is more balanced. We followed this construction so that we could better compare ERM vs. reweighting vs. group DRO techniques using a stable set of hyperparameters. In practice, if the validation set were also skewed, we might expect hyperparameter tuning based on worst-group accuracy to be more challenging and noisy.

Due to the above procedure, when reporting average test accuracy in our experiments, we calculate the average test accuracy over each group and then report a weighted average, with weights corresponding to the relative proportion of each group in the (skewed) training dataset.

**CelebA.** We use the official train-val-test split that accompanies the CelebA celebrity face dataset (Liu et al., 2015). We use the *Blond_Hair* attribute as the target label and the *Male* attribute as the spuriously-associated variable.

## C.2 MODELS

**ResNet50.** We use the Pytorch `torchvision` implementation of the ResNet50 model, starting from pretrained weights.

We train the ResNet50 models using stochastic gradient descent with a momentum term of $0.9$ and a batch size of $128$; the original paper used batch sizes of $128$ or $256$ depending on the dataset (He et al., 2016). As in the original paper, we used batch normalization (Ioffe & Szegedy, 2015) and no dropout (Srivastava et al., 2014). For simplicity, we train all models without data augmentation.

We use a fixed learning rate instead of the standard adaptive learning rate schedule to make our different model types easier to directly compare, since we expected the scheduler to interact differently with different model types (e.g., due to the different definition of loss). The interaction between batch norm and $\ell_2$ penalties means that we had to adjust learning rates for each different $\ell_2$ penalty strength (and each dataset). The learning rates below were chosen to be the highest learning rates that still resulted in stable optimization.

For the standard training experiments in Section 3.1, we use a $\ell_2$ penalty of $\lambda = 0.0001$ (as in He et al. (2016)) for both Waterbirds and CelebA, with a learning rate of $0.001$ for Waterbirds and

0.0001 for CelebA. We train the CelebA models for 50 epochs and the Waterbirds models for 300 epochs.

For the early stopping experiments in Section 3.2, we train each ResNet50 model for 1 epoch. For the strong $\ell_2$ penalty experiments in that section, we use $\lambda = 1.0$ for Waterbirds and $\lambda = 0.1$ for CelebA, with both datasets using a learning rate of 0.00001. These settings of $\lambda$ differ because we found that the lower value was sufficient for controlling overfitting on CelebA but not on Waterbirds.

For the group adjustment experiments in Section 3.3, we use the same settings of $\lambda = 1.0$ for Waterbirds and $\lambda = 0.1$ for CelebA, with both datasets using a learning rate of 0.00001. For both datasets, we use the value of $C \in \{0, 1, 2, 3, 4, 5\}$ found in the benchmark grid search described below.

For the benchmark in Section 4 (Table 3), we grid search over $\ell_2$ penalties of $\lambda \in \{0.0001, 0.1, 1.0\}$ for Waterbirds and $\lambda \in \{0.0001, 0.01, 0.1\}$ for CelebA, using the corresponding learning rates for each $\lambda$ and dataset listed above. (Waterbirds and CelebA at $\lambda = 0.1$, which is not listed above, both use a learning rate of 0.0001.) To avoid advantaging DRO by allowing it to try many more hyperparameters, we only test group adjustments (searching over $C \in \{0, 1, 2, 3, 4, 5\}$) on the $\ell_2$ penalties used in Section 3.3, i.e., $\lambda = 1.0$ for Waterbirds and $\lambda = 0.1$ for CelebA. All benchmark models were evaluated at the best early stopping epoch (as measured by robust validation accuracy).

**BERT.** We use the Hugging Face `pytorch-transformers` implementation of the BERT `bert-base-uncased` model, starting from pretrained weights (Devlin et al., 2019).[5] We use the default tokenizer and model settings from that implementation, including a fixed linearly-decaying learning rate starting at 0.00002, AdamW optimizer, dropout, and no $\ell_2$ penalty ($\lambda = 0$), except that we use a batch size of 32 (as in Devlin et al. (2019)) instead of 8. We found that this slightly improved robust accuracy across all models and made the optimization less noisy, especially on the ERM model.

For the standard training experiments in Section 3.1, we train for 20 epochs.

For the $\ell_2$ penalty experiments in Section 3.2, we tried penalties of $\lambda \in \{0.01, 0.03, 0.1, 0.3, 1.0, 3.0, 10.0\}$. However, these models had similar or worse robust accuracies compared to the default BERT model with no $\ell_2$ penalty.

For the early stopping experiments in Section 3.2, we train for 3 epochs, which is the suggested early-stopping time in Devlin et al. (2019).

For the benchmark in Section 4 (Table 3), we similarly trained for 3 epochs. All benchmark models were evaluated at the best early stopping epoch (as measured by robust validation accuracy).

---

[5]https://github.com/huggingface/pytorch-transformers

