# OpenReview forum: "Distributionally Robust Neural Networks"
_ICLR.cc/2020/Conference — Accept (Poster)_

### Official Review · AnonReviewer2 · 2019-10-20
**Official Blind Review #2**

**Rating:** 3

**Review:**

This paper proposes a novel training method based on group DRO that makes it possible for overparameterized neural networks to achieve uniform accuracy over different groups of the data, where the data is required to obey a mixture distribution composed of these groups. In addition, the authors identify that increased regularization plays a critical role for worst-group performance in the overparameterized regime by a series of empirical studies. The authors also compare  their method with the traditional importance weighting method. Finally, they introduce a stochastic optimizer for their group DRO method.

Pros:

* The experiments are well designed and the results show the effectiveness of the approach.
* Most parts of the paper are well written and easy to follow.

 Cons:
* The title is a little bit vague and may overstate the paper's contribution. In fact, the problem addressed in the paper is not as general as the title suggests and has little to do with the generic DRO framework.
* In the group DRO setting the data is distributed as a mixture of different groups. However, the groups need to be chosen using prior knowledge of "spurious associations", which may arouse doubts about the algorithm's actual effectiveness and undermines the value of the algorithm on the application side.
* Some terms used in the paper are ambiguous. For example the concept of  "group" is not defined properly. Sometimes it refers to a subset of the dataset while sometimes it denotes a sub-distribution in the mixture distribution. The authors should use more precise terminology.
* There is a typo in Proposition 1, where the uncertainty set Q should be a set of probability distributions,  not a subset of R^m.

**Experience Assessment:**

I have read many papers in this area.

**Review Assessment: Checking Correctness Of Derivations And Theory:**

I assessed the sensibility of the derivations and theory.

**Review Assessment: Checking Correctness Of Experiments:**

I assessed the sensibility of the experiments.

**Review Assessment: Thoroughness In Paper Reading:**

I read the paper at least twice and used my best judgement in assessing the paper.

---

> ### Author Response · Authors · 2019-11-13
> **Response to Reviewer #2**
>
> Thank you very much for your detailed and helpful comments.
>
>
> 1. “The title is a little bit vague and may overstate the paper's contribution. In fact, the problem addressed in the paper is not as general as the title suggests and has little to do with the generic DRO framework.”
>
> We will change our title to “Distributionally Robust Neural Networks for Group Shifts: On the Importance of Regularization for Worst-Case Generalization.” We hope the new title highlights our focus on the group DRO setting as well as our specific contributions. In addition, we will edit the main text to further emphasize that our work is on the group DRO setting.
>
>
> 2. “The groups need to be chosen using prior knowledge of ‘spurious associations’, which may arouse doubts about the algorithm's actual effectiveness and undermines the value of the algorithm on the application side.”
>
> Please refer to our response to all reviewers. This is an important point that we will further discuss in the updated manuscript.
>
>
> 3. “Some terms used in the paper are ambiguous. For example the concept of "group" is not defined properly. Sometimes it refers to a subset of the dataset while sometimes it denotes a sub-distribution in the mixture distribution. The authors should use more precise terminology.”
>
> By a “group”, we refer to a sub-distribution in a mixture distribution. If the mixture distribution is a population distribution, then the group refers to a sub-distribution of the population distribution; and if the mixture distribution is an empirical distribution (e.g., the set of training examples sampled from the population), then the group refers to sub-distribution of the empirical distribution, which corresponds to a subset of the sampled dataset. We will define groups more carefully in Section 2 and distinguish between the empirical and population distributions in other sections. If there are any specific parts in which such clarifications would be helpful, please let us know.
>
>
> 4. “There is a typo in Proposition 1, where the uncertainty set Q should be a set of probability distributions, not a subset of R^m.”
>
> That is indeed a typo, and we will fix it as suggested. Thank you.

---

### Official Review · AnonReviewer1 · 2019-10-24
**Official Blind Review #1**

**Rating:** 8

**Review:**

This paper describes a method of training neural networks to be robust to a worse case mixture of a set of predefined example attributes. This is done with a loss in accuracy in the average case but improvements in the worse case. The proposed algorithm is relatively simple and convergence rates are also given for this new algorithm.

Building neural networks that perform well in the face of group-level worse case test-set distributions is a very important problem particularly in areas such as health and safety-critical applications as past work points out. This paper shows good results in the worse case and additionally shows that the common technique of importance reweighting cannot arrive at the same solution. The convergence analyses also yield additional insight into this new algorithm. The paper is well written and relatively easy to understand with good details on the experimental setup. The algorithm has a downside in that the groups must be known a priori, is it possible for these groups to be learnt? Also, can using a hinge loss also improve robustness to the worse case examples?

However, there are some unanswered questions in the paper. What is the effect on the training time of this algorithm? Is it just the time for an additional forward prop? What is the effect on the worse-case examples of weight decay on the Bert model? Even though it hurts the average performance does it improve the worse case at all? In the last line of algorithm 1 why is q_G used instead of q_g?

Other comments:
At the bottom of P5: The ordering of 93.4% and 97.1% seem to be reversed.
Above eq. 5 , \delta_g seems to be overloaded. In the paragraph, it first refers to the generalization gap and then later to a heuristic.
Table 2: The drop in average accuracy for waterbirds does not seem 'small'.
Bottom of P8: 'background is more unique', it seems this is supposed to mean the background appears less often?

======================================================================================================
Update after rebuttal:

Thanks for the detailed answers to my comments and the additional experiments done with a hinge loss. I will keep my rating.

**Experience Assessment:**

I have read many papers in this area.

**Review Assessment: Checking Correctness Of Derivations And Theory:**

I assessed the sensibility of the derivations and theory.

**Review Assessment: Checking Correctness Of Experiments:**

I assessed the sensibility of the experiments.

**Review Assessment: Thoroughness In Paper Reading:**

I read the paper at least twice and used my best judgement in assessing the paper.

---

> ### Author Response · Authors · 2019-11-13
> **Response to Reviewer #1**
>
> Thank you very much for your detailed comments and for going through our paper so carefully.
>
>
> 1. “The algorithm has a downside in that the groups must be known a priori, is it possible for these groups to be learnt?”
>
> Thanks for raising this point; please refer to our response to all reviewers.
>
>
> 2. “Can using a hinge loss also improve robustness to the worse case examples?”
>
> We ran new experiments using the hinge loss in the CelebA and Waterbirds datasets (since these are binary classification tasks), but performance on the worst-case group did not improve. Compared to the logistic loss, performance with the hinge loss was similar or slightly worse across each group. For CelebA (on the test set and with early stopping), robust accuracy was 84.4% with the hinge loss and 88.3% with the logistic loss; and similarly for Waterbirds, robust accuracy was 84.7% with the hinge loss and 84.9% with the logistic loss.
>
>
> 3. “What is the effect on the training time of this algorithm? Is it just the time for an additional forward prop?”
>
> Optimizing for the worst-case group loss instead of the average loss has little effect on the run time of the algorithm. In practice, optimizing for the worst-case group loss vs. average loss takes a similar amount of time (<5% difference) for a given number of epochs. For example, it took 12h 50min to train the CelebA ERM model for 50 epochs on an NVIDIA TITAN Xp, and 13h 20 min to train the corresponding DRO model.
>
> The small difference above is expected because the computation of the loss $\ell(\theta^{t-1}; (x,y))$ and its gradient $\nabla\ell(\theta^{t-1}; (x,y))$ dominates the run time of each iteration in both optimization algorithms. The robust optimizer in Algorithm 1 requires only a few additional computations over the standard optimizer: a. Multiplying the weights $q$ by exponentiated losses and normalizing (second-to-last line in Algorithm 1), and b. Multiplying the loss gradient by $q$ (last line in Algorithm 1). These are relatively cheap operations.
>
> This comparison of training times is an important point to clarify in the paper, and we will make it explicit in our revision. Thanks for highlighting it.
>
>
> 4. “What is the effect on the worse-case examples of weight decay on the BERT model? Even though it hurts the average performance does it improve the worse case at all?
>
> On the BERT model, we found that weight decay (specifically, $\lambda \in \{0.01, 0.03, 0.1, 0.3, 1.0\}$) did not seem to affect the model’s accuracy on the worst-case group in comparison to the model without weight decay; training the above models to convergence resulted in similar (bad) performance on the worst-case group.
>
> Much larger values of weight decay ($\lambda \in \{3.0, 10.0\}$) were too conservative and significantly lowered model performance across all groups. In CelebA and Waterbirds datasets, we similarly observed that weight decays that are too high result in poor overall performance.
>
> While it is possible that we have not found the appropriate value of weight decay for the BERT/MultiNLI model, these results may suggest that weight decay is not an effective form of regularization (compared to early stopping) for this particular model.
>
>
> 5. “In the last line of algorithm 1 why is $q_G$ used instead of $q_g$?”
>
> That is a typo, and it should indeed be $q_g$. We will fix it. Thank you.
>
>
> 6. “At the bottom of P5: The ordering of 93.4% and 97.1% seem to be reversed.”
>
> That is also a typo, and we will fix it. Thank you.
>
>
> 7. “Above eq. 5 , $\delta_g$ seems to be overloaded. In the paragraph, it first refers to the generalization gap and then later to a heuristic.”
>
> Thanks for catching the overloading. We will refer to the heuristic as $\hat{\delta}_g$ instead.
>
>
> 8. “Table 2: The drop in average accuracy for waterbirds does not seem 'small'.”
>
> You’re right that it is not a small change in relative error. Thanks for the catch; we will clarify the text.
>
>
> 9. Bottom of P8: 'background is more unique', it seems this is supposed to mean the background appears less often?
>
> We will explain the example more clearly in the updated version. The idea is the following: Assume that the actual spurious association is the same as in the original Waterbirds dataset (whether the background is water or land). However, instead of a single water background and single land background, we now have fine-grained labelings of water and land backgrounds, such that waterbirds appear in 9 different water backgrounds (e.g., “lake”, “pond”, “sea”, etc.) and 1 land background, while landbirds appear in 9 different land backgrounds and 1 water background. In this setting, each group of (waterbird/landbird, background) is the same size, so resampling yields the same model as ERM. However, the DRO model would correctly upweight the waterbirds on a land background (and vice versa).

---

### Official Review · AnonReviewer5 · 2019-11-05
**Official Blind Review #5**

**Rating:** 6

**Review:**

To the best of my knowledge, this is the first paper to carefully address and propose an algorithm (with guarantees) for distributionally robust learning in the overparametrized regime, which is typical of modern large deep neural networks. Following other work, the paper formalizes distributionally robust learning as the minimization of a worst-case loss over a set of possible distributions. The main message of the paper is that for distributionally robust learning, regularization plays an important role by avoiding perfect fitting to the training data, at the cost of poor generalization (thus lack of robustness) in some of the possible distributions. Furthermore, the paper proposes a new stochastic optimization algorithm to minimize the loss that corresponds to distributionally robust learning and gives convergence guarantees for the proposed algorithm.

I think this is an interesting and solid paper, with a clear presentation style, and well-supported contributions. To the best of my knowledge, this is novel work and, in my opinion, it is relevant work, both in terms of applicability as well as in terms of contribution to the understanding of the generalization behavior of overparameterized deep neural networks.


**Experience Assessment:**

I have read many papers in this area.

**Review Assessment: Checking Correctness Of Derivations And Theory:**

I did not assess the derivations or theory.

**Review Assessment: Checking Correctness Of Experiments:**

I assessed the sensibility of the experiments.

**Review Assessment: Thoroughness In Paper Reading:**

I read the paper at least twice and used my best judgement in assessing the paper.

---

> ### Author Response · Authors · 2019-11-13
> **Response to Reviewer #5**
>
> Thank you very much for your comments. We appreciate the emergency review!

---

### Author Response · Authors · 2019-11-13
**Response to all reviewers**

We thank all of our reviewers for their detailed and very helpful comments. We have responded to each reviewer individually and will update our paper with the corresponding changes.

Here, we discuss Reviewers 1 and 2's comment about how our method uses pre-specified groups. Prior work has shown that it is difficult to learn distributionally robust models without some prior knowledge or structural assumptions on groups. Without any assumptions, the worst-case group of training data could be exactly the group of training points with the highest loss, but minimizing the loss on these worst-case points often yields models that are too conservative to be effective (Hu et al., 2018; Oren et al., 2019). Asking domain experts to pre-specify groups is a direct way of incorporating such prior knowledge, and as we discuss below, we believe it is feasible in many applications. As Reviewer 1 suggests, exploring other approaches that build upon the group DRO framework (e.g., learning groups by making assumptions on group structure) would be promising future work.

Existing work in many application areas has identified realistic distributional shifts that can be practically expressed in terms of pre-specified groups. For instance, in biology, batch effects like the day on which an experiment was performed are a significant source of spurious variability (e.g., Leek et al., 2010); and in medicine, patient demographics like ethnicity are spuriously associated with patient outcomes (e.g., Badgeley et al., 2018). In NLP, we studied a specific annotation artifact (negations), but recent work has categorized many others (Gururangan et al., 2018; Naik et al., 2018; Liu et al., 2019; McCoy et al., 2019). In all of these examples, the group identity of each training example is already present in the dataset, and we can use it to build models that are robust to the corresponding distributional shifts.

Moreover, we need not be too precise when specifying groups. Concretely, it can be sufficient to specify a set of candidate attributes for spurious correlations, as long as one is related to the ground-truth spurious attribute. In a new CelebA experiment, group DRO maintains high robust accuracy when: 1) instead of the ground-truth spurious attribute Male, we specify a related attribute Wearing_Lipstick instead, and 2) we also specify four distractor/non-spurious attributes (Eyeglasses, Smiling, Double_Chin, and Oval_Face). Optimizing for worst-case performance over all $2^6 = 64$ groups, the DRO model attains 78.9% robust accuracy across the 4 original groups (dark-haired males and females, and blond males and females). These robust accuracies are not far off from the original DRO model with just the ground-truth spurious attribute (86.7%) and significantly outperform the ERM model (37.8%).

Of course, there are still many applications that do not lend themselves to pre-specified groups. We agree with Reviewer 1 that learning groups is an interesting direction for future work; the key challenge there is to understand what structural assumptions on groups (e.g., we might assume data points within a group cluster together) lead to meaningful groups in practice.


References:
1. W. Hu, G. Niu, I. Sato, and M. Sugiyama. Does distributionally robust supervised learning give robust classifiers? In International Conference on Machine Learning (ICML), 2018.
2. Y. Oren, S. Sagawa, T. Hashimoto, and P. Liang. Distributionally robust language modeling. In Empirical Methods in Natural Language Processing (EMNLP), 2019
3. R. T. McCoy, E. Pavlick, and T. Linzen. Right for the wrong reasons: Diagnosing syntactic heuristics in natural language inference. In Association for Computational Linguistics (ACL), 2019.
4. S. Gururangan, S. Swayamdipta, O. Levy, R. Schwartz, S. Bowman, and N. A. Smith. Annotation artifacts in natural language inference data. In Association for Computational Linguistics (ACL), 5. 2018.
5. A. Naik, A. Ravichander, N. Sadeh, C. Rose, and G. Neubig. Stress test evaluation for natural language inference. In International Conference on Computational Linguistics (COLING), pp. 2340–2353, 2018.
6. N. F. Liu, R. Schwartz, and N. A. Smith. Inoculation by Fine-Tuning: A Method for Analyzing Challenge Datasets. In North American Chapter of the Association for Computational Linguistics (NAACL), 2019.
7. M.A. Badgeley, J.R. Zech, L. Oakden-Rayner, B.S .Glicksberg, M. Liu, W. Gale, M.V. McConnell, B. Percha, T. M. Snyder, and J. T. Dudley. Deep learning predicts hip fracture using confounding patient and healthcare variables. npj Digital Medicine, 2, 2019.
8. J. T. Leek, R. B. Scharpf, H. C. Bravo, D. Simcha, B. Langmead, W. E. Johnson, D. Geman, K. Baggerly, and R. A. Irizarry. Tackling the widespread and critical impact of batch effects in high-throughput data. Nature Reviews Genetics, 11(10), 2010.

---

### Decision · Program_Chairs · 2019-12-19

**Decision:**

Accept (Poster)

**Comment:**

This paper proposes distributionally robust optimization (DRO) to learn robust models that minimize worst-case training loss over a set of pre-defined groups. They find that increased regularization is necessary for worst-group performance in the overparametrized regime (something that is not needed for non-robust average performance).

This is an interesting paper and I recommend acceptance. The discussion phase suggested a change in the title which slightly overstated the paper's contributions (a comment which I agree with). The authors agreed to change the title in the final version.